# Prenatal Molecular Hydrogen Administration Ameliorates Several Findings in Nitrofen-Induced Congenital Diaphragmatic Hernia

**DOI:** 10.3390/ijms22179500

**Published:** 2021-08-31

**Authors:** Mayo Miura, Kenji Imai, Hiroyuki Tsuda, Rika Miki, Sho Tano, Yumiko Ito, Shima Hirako-Takamura, Yoshinori Moriyama, Takafumi Ushida, Yukako Iitani, Tomoko Nakano-Kobayashi, Shinya Toyokuni, Hiroaki Kajiyama, Tomomi Kotani

**Affiliations:** 1Department of Obstetrics and Gynecology, Nagoya University Graduate School of Medicine, 65 Tsurumai-Cho, Showa-Ku, Nagoya 466-8550, Japan; miura-mayo@med.nagoya-u.ac.jp (M.M.); kenchan2@med.nagoya-u.ac.jp (K.I.); tano.sho@med.nagoya-u.ac.jp (S.T.); u-taka23@med.nagoya-u.ac.jp (T.U.); yukakoi@med.nagoya-u.ac.jp (Y.I.); tnakano@med.nagoya-u.ac.jp (T.N.-K.); kajiyama@med.nagoya-u.ac.jp (H.K.); 2Department of Obstetrics and Gynecology, Japanese Red Cross Nagoya First Hospital, 3-35 Michisita-Cho, Nakamura-Ku, Nagoya 453-8511, Japan; hiro-t@med.nagoya-u.ac.jp (H.T.); y-ito@nagoya-1st.jrc.or.jp (Y.I.); 3Laboratory of Bell Research Center, Department of Obstetrics and Gynecology Collaborative Research, Nagoya University Graduate School of Medicine, 65 Tsurumai-Cho, Showa-Ku, Nagoya 466-8550, Japan; rika-miki@kishokai.or.jp; 4Department of Obstetrics and Gynecology, Kasugai Municipal Hospital, Kasugai 486-8510, Japan; shimanbo.ccb@hotmail.co.jp; 5Department of Obstetrics and Gynecology, Fujita Health University Graduate School of Medicine, Toyoake 470-1192, Japan; yoshinori.moriyama@fujita-hu.ac.jp; 6Department of Pathology and Biological Responses, Nagoya University Graduate School of Medicine, 65 Tsurumai-Cho, Showa-Ku, Nagoya 466-8550, Japan; toyokuni@med.nagoya-u.ac.jp; 7Center for Maternal-Neonatal Care, Division of Perinatology, Nagoya University Hospital, 65 Tsurumai-Cho, Showa-Ku, Nagoya 466-8560, Japan

**Keywords:** congenital diaphragmatic hernia, oxidative stress, platelet-derived growth factor

## Abstract

Oxidative stress plays a pathological role in pulmonary hypoplasia and pulmonary hypertension in congenital diaphragmatic hernia (CDH). This study investigated the effect of molecular hydrogen (H_2_), an antioxidant, on CDH pathology induced by nitrofen. Sprague-Dawley rats were divided into three groups: control, CDH, and CDH + hydrogen-rich water (HW). Pregnant dams of CDH + HW pups were orally administered HW from embryonic day 10 until parturition. Gasometric evaluation and histological, immunohistochemical, and real-time polymerase chain reaction analyses were performed. Gasometric results (pH, pO_2,_ and pCO_2_ levels) were better in the CDH + HW group than in the CDH group. The CDH + HW group showed amelioration of alveolarization and pulmonary artery remodeling compared with the CDH group. Oxidative stress (8-hydroxy-2′-deoxyguanosine-positive-cell score) in the pulmonary arteries and mRNA levels of protein-containing pulmonary surfactant that protects against pulmonary collapse (surfactant protein A) were significantly attenuated in the CDH + HW group compared with the CDH group. Overall, prenatal H_2_ administration improved respiratory function by attenuating lung morphology and pulmonary artery thickening in CDH rat models. Thus, H_2_ administration in pregnant women with diagnosed fetal CDH might be a novel antenatal intervention strategy to reduce newborn mortality due to CDH.

## 1. Introduction

Congenital diaphragmatic hernia (CDH) is a life-threatening anomaly with a mortality rate as high as 20–40% [1,2]. Its prevalence is known to be 1:4000 live births [3]. CDH presents a defect in the diaphragm through which the abdominal organs migrate into the thoracic cavity. Therefore, pulmonary growth is restricted by the compression of the migrated organs into the thoracic cavity during the fetal period. Pulmonary hypoplasia is associated with pulmonary hypertension (PH) after birth. PH seen in CDH is categorized as PH due to developmental lung disorders [4], PH has been increasingly recognized as a significant factor for mortality and morbidity in newborns with CDH [5]. Extracorporeal membrane oxygenation (ECMO) has been used in patients with CDH who are unresponsive to optimal medication and ventilation. However, this therapeutic approach can only be used after birth. The pathology of PH in patients with CDH is characterized by an extensive muscularization of vessels and pulmonary vascular remodeling, which starts early during gestation [6]. Currently, a clinical trial of fetoscopic endoluminal tracheal occlusion (FETO) as a prenatal intervention for CDH is ongoing [7]. FETO has been shown to improve survival rate by increasing lung volume but has the risk of premature delivery due to fetoscopy [5,8]. Therefore, prenatal medical therapies should be explored as alternatives [9]. We and others have reported the pathological role of oxidative stress in pulmonary hypoplasia and PH in CDH [10,11,12].

Molecular hydrogen (H_2_) was first reported as an antioxidant that reduces reactive oxygen species (ROS; OH· and peroxynitrite) [13]. H_2_ has several benefits in clinical applications, including rapid diffusion and lack of toxicity. H_2_ has been reported to exert preventive and therapeutic effects on various pathologies, such as ischemia, injury, metabolic syndrome, and inflammation in different organs [14]. We have reported that maternal administration of H_2_ increases its concentration in the organs of pups and improves brain or lung damage in pups [15,16,17,18]. Another study reported that H_2_ ameliorates pulmonary artery hypertension (PH) with decreased smooth muscle cell (SMC) proliferation [19]. However, the mechanisms underlying the effects of H_2_ on the pulmonary artery remain unknown. Receptor tyrosine kinases, including platelet-derived growth factor (PDGF), have been used as therapeutic targets for PH [20] and several studies have indicated an essential role for PDGF, especially the PDGF-B chain, in PH [21,22]. PDGF-B and PDGF receptor-beta (PDGFRβ) signaling contributes to the excessive proliferation and migration of pulmonary SMCs [20,23]. PDGF-BB has a more substantial mitogenic effect on human vascular SMCs than PDGF-AA [24]. PDGF levels are reported to increase in the amniotic fluid of human CDH [25] and the lung tissues of animal CDH models [11,21]. Furthermore, PDGF-B signaling plays a pathological role in CDH and is associated with oxidative stress [11]. Although H_2_ indirectly inhibits signal transduction, the modulation of PDGF-B signaling by H_2_ remains unknown. 

Based on these findings, we aimed to investigate whether prenatal H_2_ treatment could improve pulmonary development and pulmonary artery remodeling in CDH. We also investigated the effect of H_2_ on oxidative stress in pulmonary arteries. 

## 2. Results

### 2.1. Prenatal H_2_ Administration Did Not Change Lung Weight but Improved Blood Gas Results in CDH Rats 

CDH occurrence was not significantly different between the Nitrofen (*n* = 14) and Nitrofen + hydrogen-rich water (HW, *n* = 17) groups (37.9 ± 22.4% vs. 39.4 ± 17.9%, *p* = 0.829). The lung weight to body weight (LW/BW) ratio of the pups in the CDH group was significantly lower than that in the control group (*p* < 0.01, Figure 1a), and the LW/BW ratio did not improve in the CDH + HW group (Figure 1a). The neonatal blood gasometric results are shown in Figure 1b–d. The pH and pO_2_ values of the CDH group were significantly lower than those of the control and CDH + HW groups (Figure 1b,d, *p* < 0.01). Furthermore, the pCO_2_ values in the CDH group were significantly higher than those in the control and CDH + HW groups (Figure 1c, *p* < 0.01, and *p* < 0.05, respectively). The pH values in the CDH + HW group were improved to a level similar to that in the control group (Figure 1b), but the pCO_2_ and pO_2_ values were significantly different between the CDH + HW and control groups (Figure 1c,d, *p* < 0.05, and *p* < 0.01, respectively). 

### 2.2. The Effects of Prenatal H_2_ Administration on Alveolarization and Pulmonary Artery Remodeling in CDH Rats

Reduced alveolarization and thickened pulmonary artery walls were observed in the CDH group compared to the control and CDH + HW groups (Figure 2a,c). More airspaces were observed in the CDH + HW group than in the CDH group (Figure 2a). The mean linear intercept (MLI) was used as a quantitative parameter for alveolarization. The MLI in the CDH group was significantly lower than that in the control and CDH + HW groups (Figure 2b, *p* < 0.01, and *p* < 0.05, respectively), and the MLI in the CDH + HW group was similar to that in the control group (Figure 2b). 

Thickening of the pulmonary artery walls was observed in the CDH group, which was significantly improved by H_2_ treatment (Figure 2c, shown with two-way arrows). To determine the effect of H_2_ on pulmonary artery remodeling, the medial wall thickness (MWT) was compared among the control, CDH, and CDH + HW groups. The average MWT (%) in the CDH + HW group was significantly lower than that in the CDH group (Figure 2d, *p* < 0.01) and was similar to that in the control group (Figure 2d). 

### 2.3. Prenatal H_2_ Administration Reduced Oxidative Stress but Did Not Affect PDGFRβ Expression in the Fetal Pulmonary Arterial Wall in CDH 

Oxidative stress was examined by 8-hydroxy-2′-deoxyguanosine (8-OHdG) positive cell scores in the wall of the pulmonary arteries among the control, CDH, and CDH + HW groups (Figure 3a, arrowheads). The scores of 8-OHdG positive cells in the CDH group were significantly higher than those in the control and CDH + HW groups (Figure 3b, *p* < 0.01, and *p* < 0.05, respectively). However, the score in the CDH + HW group was significantly higher than that in the control group (Figure 3b, *p* < 0.01).

PDGFRβ expression in the wall of the pulmonary arteries was investigated among the control, CDH, and CDH + HW groups (Figure 3c) by grading the intensity of immunohistochemical staining (Figure 3e). PDGFRβ expression in the CDH group was significantly higher than that in the control group (Figure 3d, *p* < 0.05). The CDH + HW group showed a decreasing trend compared with that in the CDH group, but no significant difference was detected between the CDH and CDH + HW groups (Figure 3d). 

Furthermore, a positive relationship was detected between the 8-OHdG positive cell score and MWT (Figure 3f, *p* < 0.01, *ρ* = 0.441). 

To examine the mechanisms underlying the effect of H_2_ on the MWT, the expression of vascular endothelial growth factor receptor (VEGFR) mRNA, *Flt1* (Fms related receptor tyrosine kinase 1, *Vegfr-1*) and *Kdr* (kinase insert domain receptor, *Vegfr-2*), was examined among the three groups. *Flt1* and *Kdr* mRNA levels were significantly lower in the CDH group than in the control group (Figure 4a,b, *p* < 0.01 and *p* < 0.05, respectively). *Flt1* and *Kdr* expression in the CDH + HW group was significantly higher than that in the CDH group (Figure 4a,b, *p* < 0.05), and was the same as that in the control group (Figure 4a,b). 

### 2.4. Prenatal H_2_ Administration Ameliorated Surfactant Protein A Production Insufficiency in Lungs of Pups with CDH 

To examine the mechanisms underlying the effect of H_2_ on the MLI, the mRNA expression of surfactant protein A (*Sftpa1*), B (*Sftpb*), and C (*Sf**tpc*) was compared among the three groups. The expression of *Sf**tpa1* and *Sf**tpc* was significantly lower in the CDH group than in the control group (Figure 5a and c, *p* < 0.01, and *p* < 0.05, respectively). *Sf**tpa1* expression in the CDH + HW group was significantly increased compared to that in the CDH group (*p* < 0.05), to the same level as that in the control group (Figure 5a). However, *Sf**tpc* mRNA levels in the CDH + HW group were unchanged compared to those in the CDH group (Figure 5c). There was no significant difference in *Sf**tpb* mRNA levels among the control, CDH, and CDH + HW groups (Figure 5b).

## 3. Discussion

In this study, we demonstrated the positive effect of prenatal H_2_ administration on pulmonary function in rats with CDH. Prenatal H_2_ administration improved blood gas parameters in CDH, although it did not affect CDH occurrence and pulmonary hypoplasia. Morphological analysis revealed that prenatal H_2_ administration ameliorated alveolarization and pulmonary artery remodeling in CDH rats. Immunohistochemical analyses showed that prenatal H_2_ administration reduced oxidative stress in pulmonary arterial wall cells. Moreover, prenatal H_2_ administration increased *Sf**tpa1*, *Flt1* and *Kdr* levels in the lungs of CDH rats. These results suggest that although prenatal H_2_ administration could not improve pulmonary hypoplasia, it might improve pulmonary function by enhancing alveolarization and ameliorating pulmonary remodeling. Furthermore, the reduction in oxidative stress by prenatal H_2_ administration improved pulmonary artery remodeling.

PH is the cornerstone of the CDH pathophysiology as well as pulmonary hypoplasia and is thought to be associated with SMC proliferation and pulmonary arterial remodeling [26]. Oxidative stress is a well-known cause of SMC proliferation in adult PH [27]. Recently, increased ROS have also been shown to contribute to pulmonary vascular remodeling by stimulating SMC proliferation in PH in newborns with or without CDH [28]. In the present study, oxidative damage to the pulmonary arterial wall was increased in rats with CDH. However, prenatal H_2_ administration reduces oxidative damage and attenuates pulmonary arterial remodeling in CDH. Furthermore, a positive correlation was detected between the 8-OHdG score and MWT in the pulmonary arteries. These findings suggest that oxidative damage to the pulmonary artery wall is associated with pulmonary artery remodeling in CDH. Furthermore, H_2_ has been reported to reduce the oxidative stress of pulmonary vessels and ameliorate PH in an adult mouse model of monocrotaline-induced PH [19]. H_2_ was found to suppress 8-OHdG-positive cells in the pulmonary arterial walls, which is consistent with the results of the present study. These findings suggest that a reduction of oxidative stress could also improve PH by attenuating pulmonary remodeling in CDH. 

The present study demonstrated that PDGFRβ was significantly increased in the lungs of rats with CDH, which is consistent with previous reports [21,22]. ROS from mitochondria have been reported to positively regulate PDGF signaling [29]. Furthermore, H_2_O_2_ increases PDGFRβ phosphorylation in primary neuronal cultures [30]. *N*-acetylcysteine, an antioxidant, has been reported to decrease PDGFRβ protein and ROS production in vascular SMCs [31]. However, others have reported that *N*-acetylcysteine reduces PDGF signaling in vascular SMCs without altering PDGFRβ expression [32]. These findings suggest that reducing oxidative stress results in the suppression of PDGF signaling. Prenatal H_2_ administration did not affect PDGFRβ expression in the pulmonary artery but attenuated pulmonary artery remodeling in the lungs of pups with CDH. Thus, H_2_ might attenuate pulmonary artery remodeling without affecting PDGFRβ expression. Additionally, the expression of *Flt1* and *Kdr*, as VEGFRs, was significantly decreased in CDH rats, which is consistent with previous studies [33,34,35]. These previous studies also reported that prenatal treatment with all trans retinoic acid, corticosteroids, or tracheal occlusion attenuates Flt1 and Kdr expression in nitrofen-induced rat CDH, suggesting an association between increased VEGFR expression and decreased MWT. Prenatal H_2_ administration significantly increased *Flt1* and *Kdr* expression, which might attenuate the MWT in CDH + HW. H_2_ has been reported to increase *Kdr* expression in rat lung, which is consistent with the present result [36]. Melatonin, an antioxidant, increases VEGFR-1 levels [37], which suggests that antioxidants improve VEGFR-1 expression. 

In the present study, the MLI was lower in the CDH group than in the control group, which is consistent with a previous report [38]. The reduced MLI was improved by prenatal H_2_ administration. The MLI is thought to be reduced in CDH owing to lung compression by prolapsed abdominal organs, as well as by impaired differentiation of the lung. Surfactant proteins A, B, and C, which are synthesized by alveolar epithelial cells, constitute pulmonary surfactants with surfactant phospholipids [39]. Pulmonary surfactant increases airspace by reducing the surface tension in the alveoli and plays a role in the prevention of pulmonary collapse and maintenance of a stable ventilatory capacity. *Sftpa* [40] and *Sftpc* [41] expression is reduced in the lungs of CDH rat models. A lower level of surfactant protein A from tracheal aspirates has also been reported in human CDH [42]. In the present study, insufficient *Sftpa1* expression in CDH lungs was attenuated by prenatal H_2_ administration. Thus, the recovery of *Sftpa1* expression could improve the MLI in CDH + HW. A previous report suggested that surfactant protein A mRNA expression is reduced by oxidative stress and might be reversed by antioxidants in the human pulmonary epithelial cell line [43], which is consistent with the present results.

The strengths of this study are as follows: First, this is the first study to examine the effects of H_2_ on CDH. Second, we demonstrated oxidative stress in the pulmonary arterial wall of the CDH rat model. Third, H_2_ is considered to be a safe agent for human applications [44]. Many clinical trials of HW administration in humans have been conducted previously [45,46,47]. The long-term safety of inhaled hydrogen gas in mice has already been reported [48]. Although several therapeutic and preventive effects for various obstetric diseases such as preterm labor [49], preeclampsia [50], fetal brain injury [16,51], and fetal lung injury [15] have been reported, these studies did not report short-term adverse effects. Fourth, it is already known that this method of prenatal H_2_ administration increases the hydrogen concentration in fetal lungs [15]. Fifth, prenatal H_2_ administration had an effect similar to that of vitamin D [52]. Taken together, these results suggest that prenatal H_2_ administration can be adapted to clinical settings. However, more convincing data including the duration and timing of H_2_ administration_,_ are required.

Our study had several limitations. First, due to high lethality, we could not confirm pulmonary arterial hypertension in the nitrofen-induced CDH model. Second, we studied only CDH rats. As shown in some papers, lung hypoplasia is also observed in non-CDH pups prenatally exposed to nitrofen. Therefore, it remains unclear whether pulmonary hypoplasia can be attributed to CDH pathology or nitrofen toxicity. Third, lung hypoplasia could not be improved by prenatal H_2_ treatment, even though FETO enhanced lung volume. Thus, prenatal H_2_ treatment could not be an alternative to FETO but would be effective as a combined therapy as H_2_ would mainly act on the pulmonary arteries. However, the effect of H_2_ on pulmonary artery remodeling was only shown by histological analysis in this study. Sildenafil has been reported to attenuate right ventricular hypertrophy and angiogenesis in the same CDH model [53]. Further study is required to establish the effect of H_2_ on pulmonary artery remodeling. 

## 4. Materials and Methods

### 4.1. Reagents

Nitrofen (2,4-dichlorophenyl-p-nitrophenyl ether; product no. 33374, Sigma Aldrich, Saint Louis, USA) (100 mg) was dissolved in 1 mL of olive oil and administered to animals via a gastric tube. HW was prepared using a hydrogen machine (H-DX1, Arega Co., Komaki, Japan). Its concentration was approximately 0.6 ppm. HW had a content of >0.4 mM H_2_, as reported previously [15,16,17,18,49,50,51]. Anti-PDGFRβ antibody [Y92] (ab32570, lot no. GR212663-9) was purchased from Abcam (Cambridge, UK). The anti-8-OHdG monoclonal antibody N45.1 (lot no.017 MOG-020P, Nikken Zeil, Fukuroi, Japan) was used to quantitate oxidative DNA damage in the pulmonary artery. 

### 4.2. Animals 

The experimental procedures in this study were approved by the Animal Experiment Committee of the Nagoya University Graduate School of Medicine (approval number: 28241 at 11 March 2016, 29306 at 10 March 2017, 31380 at 19 March 2018, 30282 at 22 March 2019, 20285 at 13 March 2020, M210726-001 at 23 March 2021), which complied with the ARRIVE guidelines and were carried out according to the National Institutes of Health Guide for the Care and Use of Laboratory Animals. All pregnant Sprague-Dawley rats were purchased from Japan SLC Inc. (Hamamatsu, Japan). All rats were maintained on a 12 h light/12 h dark lighting schedule (lights on at 9:00 a.m., off at 9:00 p.m.), and allowed free access to water and standard chow. 

### 4.3. Experimental Design 

Pregnant rats were randomly divided into three groups (Figure 6): control, Nitrofen, and Nitrofen + HW. The nitrofen-induced rat CDH models were prepared as previously reported [54,55]. One hundred milligrams of nitrofen (2,4-dichlorophenyl-p-nitrophenylether, Sigma-Aldrich, Inc., St. Louis, MO, USA) dissolved in 1 mL of olive oil was administered using a gastric tube on embryonic day 9 (E9) in the Nitrofen and Nitrofen + HW groups. In the control group, 1 mL of olive oil was administered on E9. In the Nitrofen + HW group, the rats were orally administered HW from E10 until parturition. Subsequently, HW was provided ad libitum, and the rats drank approximately 120 mL/kg of HW per day, as previously described [15]. HW was aliquoted into glass drinking bottles, which were changed every 24 h to prevent H_2_ degassing and air refilling, as previously reported [15,16,17,18,49,50,51]. 

### 4.4. Tissue Collection

Lung tissues were collected as described previously [54]. On E21, rats were subjected to cesarean section under anesthesia. All fetuses were investigated for the lack of a diaphragm by thoracolaparotomy. The incidence of CDH was compared between the Nitrofen and Nitrofen + HW groups. Lung samples with a diagnosis of CDH were used for subsequent analyses in both the Nitrofen and Nitrofen + HW groups. The three groups were reconstructed as follows: control; CDH, CDH pups in the Nitrofen group; and CDH + HW, CDH pups in the Nitrofen + HW group (Figure 6). 

Fetal lungs from each group were weighed to calculate the LW/BW ratio. To minimize variability, we excluded the top and bottom two values for each group. The sample size of each group is as follows: control, *n* = 63; CDH, *n* = 40; and CDH + HW, *n* = 76. The lungs were fixed by tracheal instillation of 4% paraformaldehyde, immersed overnight, and paraffin-embedded for immunohistochemical analysis. After slicing the paraffin-embedded lungs, paraffin sections were stretched, and three different groups of lungs, one from each of the three groups, were mounted on the same glass slide.

### 4.5. Gasometric Evaluation 

The measurements were performed as previously reported [54,56]. Pups were delivered on E21 via cesarean section. Blood was collected from the neck 5 min after birth. The pups were dissected to investigate the lack of a diaphragm. The pH, pCO_2_, and pO_2_ values were measured using an i-STAT analyzer (Fuso, Japan). The sample size of each group is as follows: control, *n* = 25; CDH, *n* = 26; and CDH + HW, *n* = 43.

### 4.6. Histological Analysis

Serial 4 µm sections were stained with hematoxylin and eosin (HE) and Elastica Van Gieson (EVG). The stained sections were analyzed using an Axiocam 506 color system (Carl Zeiss, Oberkochen, Germany). In HE-stained lung sections, lung morphology was assessed at 200× magnification (control, *n* = 12; CDH, *n* = 13, CDH + HW, *n* = 11). MLI was measured using ImageJ [57]. In EVG-stained lung sections, MWT was measured in the pulmonary arteries to quantify pulmonary artery remodeling, as previously described [53,54]. Only arteries with an external diameter ranging from 25 to 150 µm were assessed at 400× magnification (13 fetuses per group; median ten arteries [range: 3–16] per fetus). The thickness was shown as a percentage of the MWT, calculated as (2 × wall thickness/external diameter) × 100%. 

### 4.7. Immunohistochemistry 

The sections were also stained with primary antibodies against 8-OHdG (1:50) and PDGFRβ (1:50) overnight at 4 °C. Eight representative fields (median: 2–13) were photographed per pup and the average value was calculated for each pup (8-OHdG: control, *n* = 12; CDH, *n* = 12, CDH + HW, *n* = 10; PDGFRβ: control, *n* = 15; CDH, *n* = 14, CDH + HW, *n* = 13). To quantify 8-OHdG, the percentage of positive nuclei stained with 8-OHdG were scored among the arterial blood vessel cells as follows: 1, 0–5%; 2, 5–25%; 3, 25–50%; 4, 50–75%; 5, >75%. To quantify PDGFRβ, the staining intensity was classified as grade 0 to grade 3, as shown in Figure 3e. Two examiners independently performed the quantification, and the average was used for the analysis.

### 4.8. Real-Time Polymerase Chain Reaction (PCR)

Total RNA from the left lung of the pups was extracted using an RNeasy Mini kit and treated with DNase (Qiagen, Hilden, Germany). For first-strand complementary DNA (cDNA) synthesis, reverse transcription reaction with 1 μg of total RNA was carried out using a first-strand cDNA synthesis kit ReverTra Ace qPCR Master Mix (Toyobo Co., Ltd., Osaka, Japan). Real-time PCR was performed with Step One Plus (Applied Biosystems) using the Fast SYBR Green Reaction Mix (Applied Biosystems, Thermo Fisher Scientific Inc., Waltham, MA, USA). Quantification was performed by calculating the ratio to β-actin mRNA, the endogenous control, using the standard method [58].

The following primers were used: *Sftpa1* (NM_001270645.1): forward 5′-CAAACAATGGGAGTCCTCAGC-3′ and reverse 5′-CGCAATACTTGCAATGGCCT-3′; *Sftpb* (NM_138842.2): forward 5′-AGCCTGGAGCAAGCGATAC-3′ and reverse 5′-AACTTCCGGATCGTGTCCTG-3′, *Sftpc* (NM_017342.2): forward 5′-GATGGAGAGCCCACCGGATT-3′ and reverse 5′-CTGACTCATGTGAAGGCCCA-3′, *Actb* (NM_031144.3): forward 5′-CGAGTACAACCTTCTTGCAGC-3′ and reverse 5′-ATACCCACCATCACACCCTGG-3′, *Flt1* (NM_001309381.1): forward 5′-ACCCATCGGCAGACCAATAC-3′ and reverse 5′-GGTCAATCCGCTGCCTGATA-3′, *Kdr* (NM_013062.2): forward 5′-ACAGCATCACCAGCAGTCAG-3′ and reverse 5′-CAGGTCCCTGTGGATACACTTC-3′.

### 4.9. Statistical Analysis

Statistical analysis was performed using JMP Pro 15 software (Japan). The Steel Dwass test and Tukey’s post hoc test were used for multiple comparisons with non-normal and normal distributions, respectively. The relationship between 8-OHdG and MWT was estimated using Spearman’s correlation analysis for non-normal distributions. Statistical significance was set at *p* < 0.05.

## 5. Conclusions

To our knowledge, this is the first study to show the effects of prenatal H_2_ administration on CDH. Although H_2_ could not prevent the occurrence of CDH and increase lung weight, it improved pulmonary function with attenuated lung morphology and the pulmonary artery thickness in the CDH rat models. Thus, H_2_ administration in pregnant women with a diagnosis of CDH in the fetus might present a new antenatal intervention strategy to reduce neonatal mortality; however, further research is required to determine the exact molecular mechanisms underlying these effects.

## Figures and Tables

**Figure 1 ijms-22-09500-f001:**
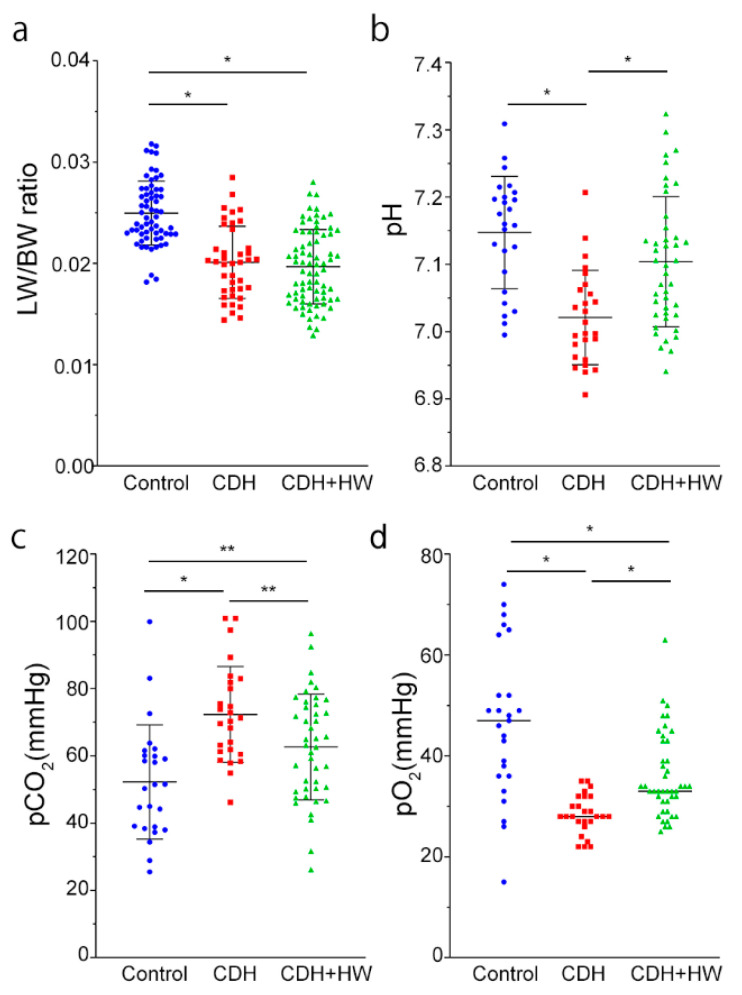
(**a**) The lung weight-to-body weight (LW/BW) ratio of control, congenital diaphragmatic hernia (CDH), and CDH + hydrogen-rich water (HW) pups. The LW/BW ratio in the CDH group was significantly lower than that in the control group, and was not improved by prenatal treatment with HW; (**b**–**d**) Neonatal blood pH (**b**), pCO_2_ (**c**), and pO_2_ (**d**) were examined. Neonatal blood was harvested 5 min after birth. Prenatal HW treatment significantly ameliorated the results of blood gas tests compared with those in the CDH group. The bars represent the mean ± standard deviation (**a**–**c**) and median (**d**) for normal and non-normal distributions, respectively. * *p* < 0.01, ** *p* < 0.05.

**Figure 2 ijms-22-09500-f002:**
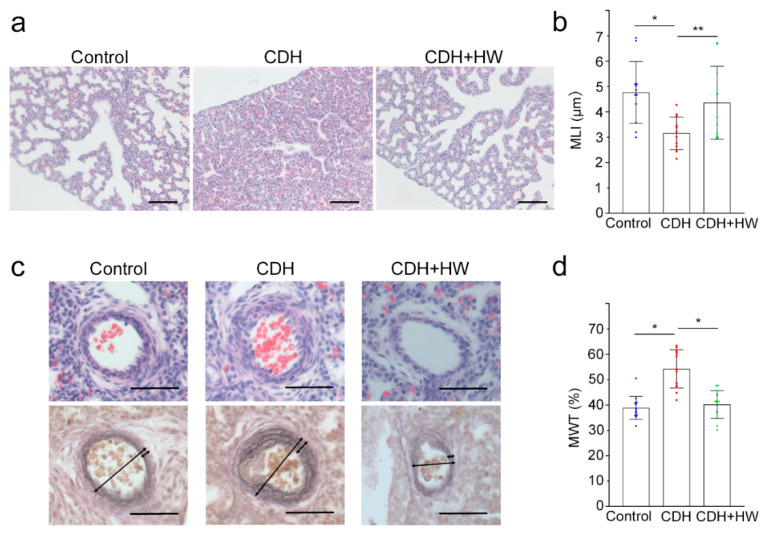
Alveolarization and pulmonary artery remodeling. (**a**) Representative images of the lungs stained with hematoxylin and eosin (HE). Original magnification, 200×. Scale bars = 100 μm; (**b**) The mean linear intercept (MLI) is shown for the control (*n* = 12), CDH (*n* = 13), and CDH + hydrogen-rich water (HW, *n* = 11) groups. Data are shown as mean ± standard deviation (SD).; (**c**) Representative images for pulmonary arteries stained with HE (upper panels) and Elastica Van Gieson (EVG, lower panels). Long and short arrows show measurements of the external diameter and wall thickness, respectively. Original magnification, 400×. Scale bars = 50 μm; (**d**) Medial wall thickness (MWT, %) was calculated as (2 × wall thickness/external diameter) × 100. MWT is shown for the control, CDH, and CDH + HW groups (each group, *n* = 13). Data are shown as mean ± SD. * *p* < 0.01, ** *p* < 0.05.

**Figure 3 ijms-22-09500-f003:**
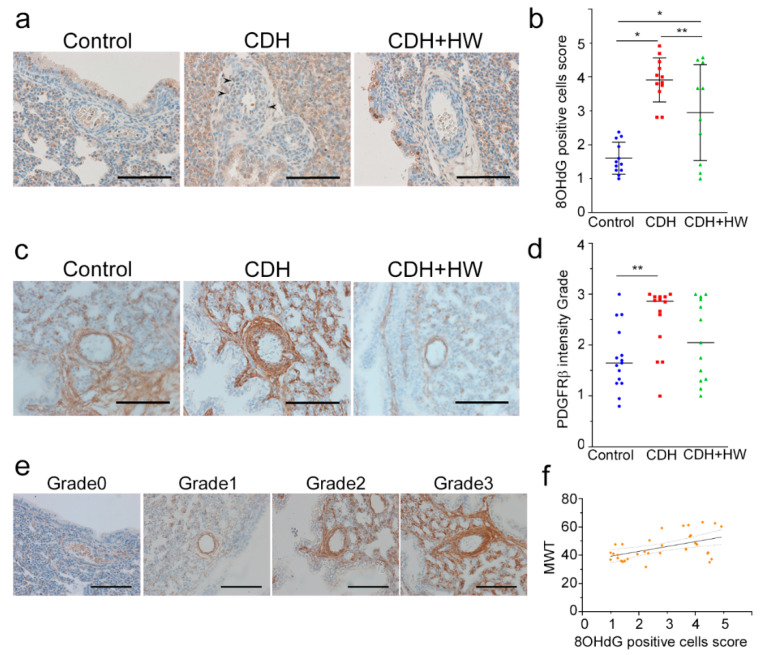
Expression of 8-hydroxy-2′-deoxyguanosine (8-OHdG) and platelet-derived growth factor receptor-beta (PDGFRβ) in pulmonary arteries. (**a**) Representative immunohistochemistry images of the pulmonary artery for 8-OHdG. The 8-OHdG positive cells are indicated by arrowheads. Original magnification, 400×. Scale bars = 100 μm; (**b**) The ratio of 8-OHdG positive cells to arterial wall cells was scored as follows, 1, 0–5%; 2, 5–25%; 3, 25–50%; 4, 50–75%; 5, 75–100%. The bars indicate the mean ± standard deviation. Pulmonary arterial walls in CDH showed significantly more 8-OHdG positive cells than in the control (*n* = 12); 8-OHdG positive cells were less in CDH+ hydrogen-rich water (HW, *n* = 10) than in CDH (*n* = 12); (**c**) Representative immunohistochemistry images of the pulmonary artery for PDGFRβ. Original magnification, 400X. Scale bars = 100 μm; (**d**) The grades of pulmonary artery immunohistochemical staining intensity for PDGFRβ are shown in the control (*n* = 15), CDH (*n* = 14), and CDH + HW (*n* = 13) groups. The bars indicate the median; (**e**) The grades 0–3 were defined as shown. Original magnification, 400X. Scale bars = 100 μm; (**f**) MWT showed a significantly positive correlation with 8-OHdG positive cells score (*p* < 0.01, *ρ* = 0.441). Values are expressed as means (solid lines), and 95% confidence bands (dotted lines). * *p* < 0.01, ** *p* < 0.05.

**Figure 4 ijms-22-09500-f004:**
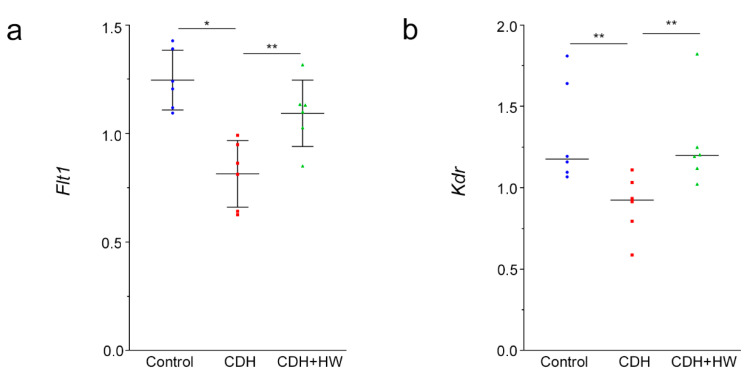
Vascular endothelial growth factor receptors (VEGFRs) mRNA expression. (**a**) *Flt1* (Fms related receptor tyrosine kinase 1, *Vegfr-1*) and (**b**) *Kdr* (kinase insert domain receptor, *Vegfr-2*) mRNA expression in the lungs derived from pups in the control, CDH, and CDH + hydrogen-rich water (HW) groups (each group, *n* = 6). The bars indicate the mean ± standard deviation (SD) (**a**) and the median (**b**) for normal and non-normal distributions, respectively. * *p* < 0.01, ** *p* < 0.05.

**Figure 5 ijms-22-09500-f005:**
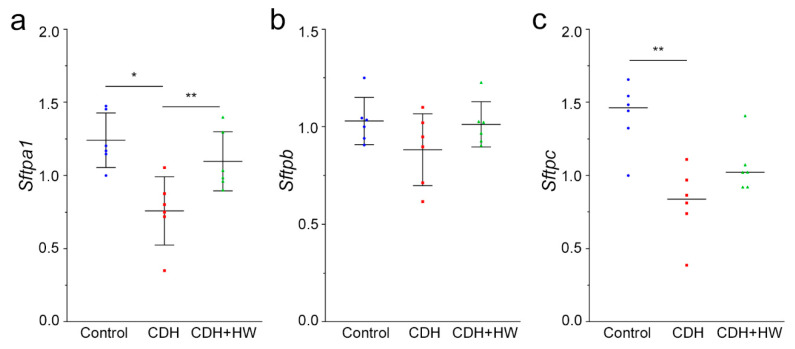
Surfactant protein A, B, C production. (**a**) Surfactant protein A1 mRNA (*Sf**tpa1*) expression in the lungs derived from pups in the control, CDH, and CDH + hydrogen-rich water (HW) groups (each group, *n* = 6). The bars indicate the mean ± standard deviation (SD). * *p* < 0.01, ** *p* < 0.05; (**b**) Surfactant protein B mRNA (*Sf**tpb*) expression in the lungs derived from pups in the control, CDH, and CDH + HW groups (each group, *n* = 6). The bars indicate the mean ± SD; (**c**) Surfactant protein C mRNA (*Sf**tpc*) expression in the lungs derived from pups in the control, CDH, and CDH + HW groups (each group, *n* = 6). The bars represent the median. ** *p* < 0.05.

**Figure 6 ijms-22-09500-f006:**
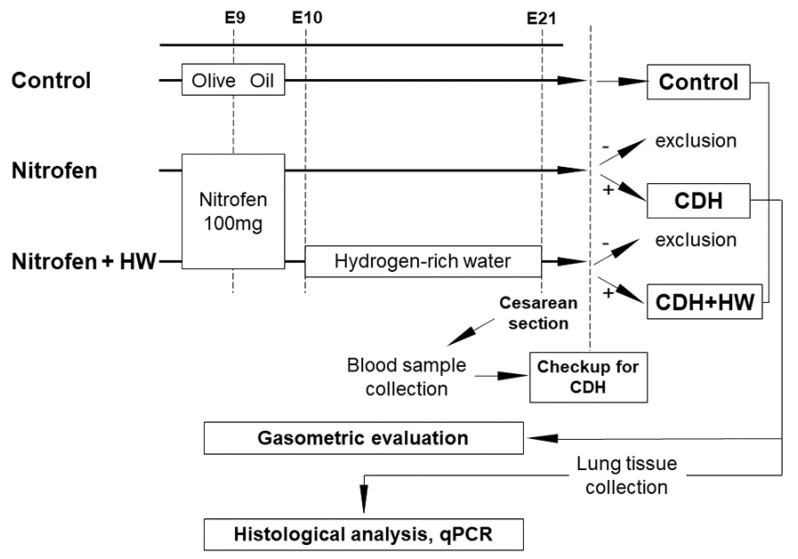
Schema of the design and protocols. Three groups were reconstructed as control, CDH, and CDH + HW. All analyses were compared among these groups. Blood samples were collected before CDH was diagnosed, but the data were only used in these specific groups. CDH, congenital diaphragmatic hernia. HW, hydrogen-rich water.

## Data Availability

The datasets used and/or analyzed during the current study are available from the corresponding author upon reasonable request.

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
