# Peer review of "Prenatal Molecular Hydrogen Administration Ameliorates Several Findings in Nitrofen-Induced Congenital Diaphragmatic Hernia"

_ijms, 2021, doi:10.3390/ijms22179500_

Round 1
Reviewer 1 Report
The title claimed: "Prenatal molecular hydrogen administration ameliorates pulmonary function in nitrofen-induced congenital diaphragmatic hernia”
Should be more appropriate change in the title “pulmonary function” with “pulmonary vascular remodeling”.
Overall, the study is well design to evaluate the final aim of the strategy.
Nevertheless, the work should be improved to have a more convincing and robust data in order to translate this strategy into the clinical scenario.
If possible, to analysis the RNA extracted from the study in order to investigate other pathways involved in the pulmonary hypertension secondary to CDH such as:
- VEGF (VEGFR-1 and VEGFR-2)
- FoxF1
- Endoglin
And also, can be performed more histological and immunofluorescence analysis such as:
- Masson's trichrome
- von Willebrand factor (vWF)
- PCNA
- anti-smooth muscle actin
- Col-1 and Col-4
Author Response
Response to Reviewer 1 comments
- The title claimed: "Prenatal molecular hydrogen administration ameliorates pulmonary function in nitrofen-induced congenital diaphragmatic hernia” Should be more appropriate change in the title “pulmonary function” with “pulmonary vascular remodeling”.
Response 1: Thank you for your comment. As you suggested, we revised the title into "Prenatal molecular hydrogen administration ameliorates pulmonary vascular remodeling in nitrofen-induced congenital diaphragmatic hernia”
- Overall, the study is well design to evaluate the final aim of the strategy. Nevertheless, the work should be improved to have a more convincing and robust data in order to translate this strategy into the clinical scenario. 
Response 2: We agree with the reviewer’s suggestion. We added the comment below (line 471-474).
Taken together, these results suggest that prenatal H2 administration can be adapted to clinical settings. However, more convincing data including the mechanisms of H2 on the CDH pathology, and the duration and timing of H2 administration, are required.
3. If possible, to analysis the RNA extracted from the study in order to investigate other pathways involved in the pulmonary hypertension secondary to CDH such as:
VEGF (VEGFR-1 and VEGFR-2), FoxF1, Endoglin
And also, can be performed more histological and immunofluorescence analysis such as:
Masson's trichrome, von Willebrand factor (vWF), PCNA, anti-smooth muscle actin, Col-1 and Col-4
Response 3: Thank you for your valuable suggestion. We would like to focus on the antioxidative effect, a significant function of molecular hydrogen in this study. Thus, in the molecules that you suggested, we compared the VEGFR-1 (Flt1) and VEGFR-2 (Kdr) expression, which is related to oxidative stress, among the three groups. We added the data in the results and also revised the Methods and Discussion accordingly. We will investigate the effect of molecular hydrogen on the other proteins and molecules in the future study. VEGFR-1 (Flt1) and VEGFR-2 (Kdr) expression decreased in the CDH group, consistent with previous studies. However, these expressions significantly increased by prenatal H2 treatment. We added the new data as Figure 4.

Reviewer 2 Report
The manuscript ijms-1300981 titled: “Prenatal molecular hydrogen administration ameliorates pulmonary function in nitrofen-induced congenital diaphragmatic hernia” reports the results of a study investigating the effect of molecular hydrogen (H2), an antioxidant, on congenital diaphragmatic hernia, considering that oxidative stress plays a pathological role in pulmonary hypoplasia and persistent pulmonary hypertension in this pathology. These results show that overall, prenatal H2 administration improved respiratory function by attenuating lung morphology and pulmonary artery thickening in CDH rat models. Thus, H2 administration in pregnant women with diagnosed fetal CDH might be a novel antenatal intervention strategy to reduce newborn mortality due to CDH.
The manuscript is well-structured and well-written providing sufficient background, although some points can be improved. Furthermore, the strengths and limitations of the study are widely discussed.
Molecular hydrogen (H2) is gaining significant attention from academic researchers and medical doctors around the world for its recently reported therapeutic potential. Although this knowledge is still in its infancy, in particular the molecular mechanisms underlying its effect, the preliminary data is intriguing, and more studies are required before we can scientifically claim any real benefit. Therefore, the research on disease models and mechanisms of action as the one reported here, along with clinical studies are particularly relevant because the high safety profile of molecular hydrogen could make it an effective strategy in some cases.
Minor revisions
The manuscript provides sufficient background, although some points can be improved.
In particular:
- In the Abstract, lines 24, 25, “Sprague-Dawley rats were divided into three groups: control, CDH induced by nitrofen, and CDH + hydrogen-rich water (HW).”, the authors should specify, as done later in the text, that CDH + hydrogen-rich water (HW) are CDH induced by nitrofen + hydrogen-rich water (HW). Or re-modulate the previous sentence, i.e. “This study investigated the effect of molecular hydrogen (H2), an antioxidant, on CDH pathology induced by nitrofen”.
- In the Abstract, lines 31-33, “The 8-hydroxy-2’-deoxyguanosine-positive-cell scores in the pulmonary arteries and surfactant protein A mRNA levels in the lungs were significantly higher and lower in the CDH group than in the control and CDH + HW groups, respectively.”, the authors should specify why they look at the 8-OHdG positive cell scores and to the mRNA levels of Sftpa1, or what is the meaning of the results.
- In the Results, section 2.4, as pointed out in the previous sentence, the authors should specify why they evaluate the surfactant proteins production in the lungs of pups with CDH. The rationale should be motivated, as done in the Discussion section.
- In the Discussion, lines 250-253, the sentence “Fifth, prenatal H2 administration had an effect like that of vitamin D [45], even though H2 was administered the day after nitrofen administration. As we reported previously, vitamin D administration was started simultaneously with that of nitrofen [45], which means pre-treatment with vitamin D is necessary for nitrofen to be effective.” Is not clear. Why pretreatment with vit D is necessary for nitrofen to be effective, and why the authors started the administration of vit D and nitrofen on different days? If they leave this sentence, more details should be added and explained. Otherwise, just specify which are the similar effects.
- In the Conclusions, the authors postulate that H2 administration in pregnant women with a diagnosis of CDH in the fetus might present a new antenatal intervention strategy to reduce neonatal mortality. Considering this deduction, it would be helpful to add in the Discussion some background about the use of H2 in pregnant women, if any.
Author Response
Response to Reviewer 2 comments
The manuscript ijms-1300981 titled: “Prenatal molecular hydrogen administration ameliorates pulmonary function in nitrofen-induced congenital diaphragmatic hernia” reports the results of a study investigating the effect of molecular hydrogen (H2), an antioxidant, on congenital diaphragmatic hernia, considering that oxidative stress plays a pathological role in pulmonary hypoplasia and persistent pulmonary hypertension in this pathology. These results show that overall, prenatal H2 administration improved respiratory function by attenuating lung morphology and pulmonary artery thickening in CDH rat models. Thus, H2 administration in pregnant women with diagnosed fetal CDH might be a novel antenatal intervention strategy to reduce newborn mortality due to CDH.
The manuscript is well-structured and well-written providing sufficient background, although some points can be improved. Furthermore, the strengths and limitations of the study are widely discussed.
Molecular hydrogen (H2) is gaining significant attention from academic researchers and medical doctors around the world for its recently reported therapeutic potential. Although this knowledge is still in its infancy, in particular the molecular mechanisms underlying its effect, the preliminary data is intriguing, and more studies are required before we can scientifically claim any real benefit. Therefore, the research on disease models and mechanisms of action as the one reported here, along with clinical studies are particularly relevant because the high safety profile of molecular hydrogen could make it an effective strategy in some cases.
Thank you for your helpful comments. We have carefully reviewed the specific comments and have thoroughly revised the manuscript as below.
Minor revisions
The manuscript provides sufficient background, although some points can be improved.
In particular:
- In the Abstract, lines 24, 25, “Sprague-Dawley rats were divided into three groups: control, CDH induced by nitrofen, and CDH + hydrogen-rich water (HW).”, the authors should specify, as done later in the text, that CDH + hydrogen-rich water (HW) are CDH induced by nitrofen + hydrogen-rich water (HW). Or re-modulate the previous sentence, i.e. “This study investigated the effect of molecular hydrogen (H2), an antioxidant, on CDH pathology induced by nitrofen”.
Response 1: As you suggested, we revised the sentence (lines 23-24) to “This study investigated the effect of molecular hydrogen (H2), an antioxidant, on CDH pathology induced by nitrofen”.
- In the Abstract, lines 31-33, “The 8-hydroxy-2’-deoxyguanosine-positive-cell scores in the pulmonary arteries and surfactant protein A mRNA levels in the lungs were significantly higher and lower in the CDH group than in the control and CDH + HW groups, respectively.”, the authors should specify why they look at the 8-OHdG positive cell scores and to the mRNA levels ofSftpa1, or what is the meaning of the results.
Response 2: As you suggested, we revised it as below (lines 30-33).
Oxidative stress (8-hydroxy-2’-deoxyguanosine-positive-cell score) in the pulmonary arteries and mRNA levels of protein-containing pulmonary surfactant that protects against pulmonary collapse (surfactant protein A) were significantly attenuated in the CDH + HW group compared with the CDH group.
- In the Results, section 2.4, as pointed out in the previous sentence, the authors should specify why they evaluate the surfactant proteins production in the lungs of pups with CDH. The rationale should be motivated, as done in the Discussion section.
Response 3: As you suggested, we added the explanation in the section 2.4 (line 201) and Discussion section (line 451-452,456).
- In the Discussion, lines 250-253, the sentence “Fifth, prenatal H2 administration had an effect like that of vitamin D [45], even though H2 was administered the day after nitrofen administration. As we reported previously, vitamin D administration was started simultaneously with that of nitrofen [45], which means pre-treatment with vitamin D is necessary for nitrofen to be effective.” Is not clear. Why pretreatment with vit D is necessary for nitrofen to be effective, and why the authors started the administration of vit D and nitrofen on different days? If they leave this sentence, more details should be added and explained. Otherwise, just specify which are the similar effects.
Response 4 : As you suggested, we did not compare the superiority between vitamin D and hydrogen. We deleted the sentence (line 469-470).
- In the Conclusions, the authors postulate that H2 administration in pregnant women with a diagnosis of CDH in the fetus might present a new antenatal intervention strategy to reduce neonatal mortality. Considering this deduction, it would be helpful to add in the Discussion some background about the use of H2 in pregnant women, if any.
Response 5. Thank you for your helpful suggestion. We are sorry to say that there is no clinical trial of the use of H2 in pregnant women. We are making a plan of a clinical trial in pregnant women to prevent preterm birth.